# The Influence of Gender in The Prognostic Impact of Diabetes mellitus in acute Pulmonary Embolism

**DOI:** 10.3390/jcm9113511

**Published:** 2020-10-30

**Authors:** Diana Oliveira, Teresa Brito, Catarina Elias, Marta Carreira, Mariana Serino, Inês Guerreiro, Helena Magalhães, Sara Coelho, Sara Ferreira, Emanuel Araújo, Ana Ribeiro, Patrícia Lourenço

**Affiliations:** 1Serviço de Medicina Interna do Centro Hospitalar e Universitário de São João, Alameda Prof Hernâni Monteiro, 4200-319 Porto, Portugal; mtapbrito@gmail.com (T.B.); catarina.elias@live.com.pt (C.E.); marta.soares.c@gmail.com (M.C.); saraalexandra525@gmail.com (S.F.); anamfr@gmail.com (A.R.); pamlourenco@yahoo.com (P.L.); 2Serviço de Pneumologia do Centro Hospitalar e Universitário de São João, Alameda Prof Hernâni Monteiro, 4200-319 Porto, Portugal; mariana.serino@gmail.com; 3Serviço de Oncologia do Centro Hospitalar Universitário Lisboa Central, Alameda de Santo António dos Capuchos, 1169–050 Lisboa, Portugal; ines.m.guerreiro@gmail.com; 4Serviço de Oncologia da Unidade de Saúde Local de Matosinhos, Rua Dr. Eduardo Torres, 4464-513 Matosinhos, Portugal; hmmagalhaes88@gmail.com; 5Serviço de Oncologia do Instituto Português de Oncologia do Porto Francisco Gentil, R. Dr. António Bernardino de Almeida 62, 4200-072 Porto, Portugal; saramarinacoelho@gmail.com; 6Unidade de Cuidados Agudos Polivalente do Centro Hospitalar de Leiria, R. de Santo André, 2410-197 Leiria, Portugal; e_filipe_araujo@hotmail.com; 7Faculdade de Medicina da Universidade do Porto, Alameda Prof. Hernâni Monteiro, 4200-319 Porto, Portugal

**Keywords:** diabetes mellitus, acute pulmonary embolism, gender related differences, prognostic impact, mortality

## Abstract

Diabetes mellitus (DM) predicts ominous outcomes in acute pulmonary embolism (PE). The influence of gender on the prognostic impact of DM in PE is unknown. We did a retrospective analysis of a cohort of patients hospitalized with PE between 2006 and 2013. The exclusion criteria were age <18, non-pulmonary veins thromboembolism, recurrent PE, chronic thromboembolic pulmonary hypertension, no radiologic confirmation of PE, and active neoplasia. The primary endpoint was all-cause mortality. The follow-up was from diagnosis until October 2017. We assessed the prognostic impact of DM using a multivariate Cox regression analysis. The analysis was stratified according to gender. The interaction between gender and DM in the outcome of patients with PE was tested. We studied 577 PE patients (median age 65 years, 36.9% men, 19.8% diabetic). The genders were similar regarding the prevalence of DM, the extension and location of PE, and the thrombolytic therapy or brain natriuretic peptide (BNP) value. Diabetics presented higher all-cause mortality (Hazard ratio (HR) = 2.33 (95% confidence Interval (CI) 1.513.61)) when compared with non-diabetics. However, when analysis was stratified according to gender, DM was independently associated with a worse prognosis only in women (HR = 2.31 (95% CI 1.453.65)), while in men the HR was 1.10 (95% CI 0.592.04). The interaction between gender and DM was significant (*p* = 0.04). Gender influences the prognostic impact of DM in acute PE. Diabetic women with PE have twice the long-term mortality risk, while DM is not mortality-associated in men.

## 1. Introduction

Diabetes mellitus (DM) is a very prevalent chronic disease that affects over 20 million people worldwide. It was the seventh leading cause of death in 2016 according to the World Health Organization (WHO), and is responsible for 1.6 million deaths each year. DM is a recognized factor in ominous outcomes in many cardiovascular settings [1,2].

Diabetic patients have known gender-related differences. Although premenopausal females appear to be protected against the disease in part due to the presence of estrogens, when present, the burden of DM is greater among women [3,4]. Both physiological particularities and gender differences concerning the care provided are possible explanations for this differential impact of DM according to gender in the outcome of many diseases [4,5]. It has been reported that women with DM exhibit a greater risk of coronary heart disease (CHD) and stroke than men with DM [6,7]. Diabetic women also appear to have higher cardiac mortality and all-cause mortality compared to diabetic men [3,4,7,8]. The coexistence of DM conferred a higher risk of all-cause and cardiovascular mortality upon women with heart failure with preserved ejection fraction (HFpEF), but not upon men [9,10]. Moreover, this women-selective survival disadvantage of diabetes also seems to occur in CHD [1,5].

Besides cardiovascular diseases (CVD), diabetic women are also at a greater risk of renal disease [11] and malignancy [12].

Venous thromboembolism (VTE) is the third most frequent acute cardiovascular syndrome, preceded only by myocardial infarction (MI) and stroke. The annual incidence rates of PE range from 39 to 115 per 100,000 of the population, with a rising tendency over recent years [13]. PE patients with DM have been shown to have higher mortality than non-diabetics [14]; however, a gender difference in the prognostic impact of DM in PE has never been reported. 

The aim of our study was to investigate the role of gender in the prognostic impact of DM in acute PE. Our hypothesis was that, like in other cardiovascular diseases, diabetic women would be at higher risk than men.

## 2. Material and Methods

We conducted a retrospective study in patients hospitalized with acute PE in Centro Hospitalar Universitário São João between 2006 and 2013. All patients with the international classification of diseases (ICD) 9 discharge diagnosis code of 415.1, corresponding to venous thromboembolism, were eligible for study inclusion. Patients of surgical and medical wards and patients admitted in intermediate and intensive care units were included in the study. All patients had acute PE confirmed by computed tomography pulmonary angiography (CTPA) or by high-probability nuclear ventilation–perfusion (V/Q) imaging. Only patients with a single PE episode were included. The exclusion criteria were patients <18 years old (pediatric ward), patients with venous thromboembolism in veins other than pulmonary veins, patients with chronic thromboembolic pulmonary hypertension, patients with evidence of organized non-acute pulmonary thrombus on CTPA, and patients with concomitant active neoplasia. Patients in which the code was attributed based on clinical suspicion with no imagological confirmation were also excluded. 

Demographic data, comorbidities and laboratory data were extracted from patients’ electronic files. DM was defined as either a known previous diagnosis or the current prescription of either an oral hypoglycemic agent or insulin. Other comorbidities were defined as follows. Arterial hypertension was defined as the presence of a previous diagnosis or record of antihypertensive pharmacological treatment. A hemoglobin level <13 g/dL in men and <12 g/dL in women was considered anemia. Renal dysfunction was considered when plasma creatinine was >1.5 mg/dL. Lymphopenia was considered when lymphocyte counts were <1500/µL.

The endpoint under analysis was all-cause mortality. Patients were followed from the diagnosis until October 2017. Vital status was ascertained by consulting hospital registries and by telephone contact with the patients or their relatives. When no information was obtained, we consulted the Registo Nacional de Utentes (RNU) platform.

The study protocol conforms to the ethical guidelines of the declaration of Helsinki. The retrospective design did not allow for obtaining patients’ signed informed consent.

Statistical analysis:

Categorical variables are presented as counts and proportions and continuous variables are presented as mean (standard deviation). Median (interquartile range) was used to describe variables with a highly skewed distribution. Patients’ characteristics were compared according to gender. A chi square test was used to compare categorical variables, student’s *T* test was used to compare normally distributed variables and the Man–Whitney *U* test was used for non-normally distributed variables. 

The prognostic impact of DM in acute PE was assessed using a Cox regression analysis. A multivariate model was built based on variables differently distributed in men and women, and a *p* value < 0.05 was considered as statistically significant. The analysis was stratified according to gender. The interaction between gender and DM in the long-term acute PE prognosis was formally tested. The survival curves of PE patients with and without concomitant DM were assessed and compared using the Kaplan–Meier method. 

The *p* value considered for statistical significance was 0.05. The data were stored and analyzed using SPSS software (version 20.0, IBM corp, Armonk, NY, USA).

## 3. Results

We studied 577 patients hospitalized with acute PE with no concomitant active neoplasia. The mean patient age was 65 years; 213 (36.9%) were male and 114 (19.8%) had DM. Pulmonary thromboembolism was central and bilateral in 175 (30.4%) patients and 67 (11.6%) needed thrombolysis. Patients’ characteristics and the comparisons between men and women are depicted in Table 1. Women were older (mean age 67 vs. 62 years in men, *p* = 0.003), and they had a higher prevalence of arterial hypertension, lower hemoglobin, creatinine and CRP, and higher lymphocyte counts. No differences existed between men and women concerning the prevalence of DM, PE extension, or need of reperfusion therapy. BNP was similar in men and women. Patients were followed for a median (interquartile range (IQR)) period of 52 (12–76) months; during this period, 240 patients died (41.6%). Men presented a higher mortality than women (46.9% vs. 38.5%, *p* = 0.05). comparison between survivors and non-survivors is provided as the Appendix A.

Table 2 shows the multivariate analysis of long-term acute PE mortality predictors. Among acute PE patients, those with DM presented higher all-cause mortality with an HR of 2.33 (95% CI: 1.513.61) when compared with non-diabetics; men also showed worse prognosis (HR of 1.84 (95%CI: 1.272.68)). Association with mortality was independent of age, the coexistence of arterial hypertension, the location and extension of the PE, thrombolysis, as well as admission anemia, renal dysfunction, lymphopenia, BNP and CRP. Figure 1 depicts the Kaplan–Meier survival curves according to the coexistence of DM. When analysis was stratified according to gender, DM was an independent predictor of all-cause mortality in patients with acute PE only in women (HR = 2.31 (95%CI 1.453.65), *p* < 0.001). In men, there was no independent association between the coexistence of DM and all-cause death (HR = 1.10 (95%CI 0.592.04), *p* = 0.77). The *p* value for the interaction between DM and gender in the prognosis of acute PE was 0.04. Figure 2 depicts the Kaplan–Meier survival curves of patients with and without DM separately in women in men, and Table 3 shows the multivariate Cox regression analysis of the predictors of all-cause mortality according to gender. 

## 4. Discussion

In women hospitalized with acute PE, those with DM had a more than twofold higher risk of long-term all-cause death than their non-diabetic counterparts. DM did not influence prognosis in men with acute PE. Gender appears to interact with DM in the long-term outcome of acute PE. The differential impact of DM in men and women has already been described in various clinical contexts, namely in cardiovascular [1,6,7,15], renal [11] and neoplastic diseases [12,13,16]. Venous thromboembolism can be viewed as part of the cardiovascular disease continuum, as it shares risk factors, namely diabetes and atherosclerosis [13].

Several explanations have been proposed for this interaction between gender and DM in the prognosis of morbid conditions. Of all of the cardiovascular risk factors, the greatest weight of evidence of a female disadvantage is for diabetes. DM more than doubles the risk of CHD, but confers an additional 44% risk to women compared to men; the results for stroke are similar but somewhat less extreme, and the excess female risk is 27% [2,15]. After an embolic episode, patients with VTE are at increased risk of subsequent myocardial infarction and stroke [13]. Besides cardiovascular risk, diabetic renal disease is also different in men and women. These patients commonly have central venous lines or take erythropoiesis-stimulating agents, which are known risk factors for VTE [13]. Although men show a faster progression of diabetic nephropathy and more often need renal replacement therapy, diabetic women on chronic dialysis treatment appear to have a higher death risk than diabetic men [11]. Women with type 2 DM present higher levels of factor VII coagulant activity, factor VIII, and plasminogen activator inhibitor−1 than their male counterparts [17]. This increased pro-thrombotic risk among diabetic women may be another possible factor contributing to their higher cardiovascular risk, including VTE risk. Another main mechanism possibly explaining these sex-related differences is the different hormonal environment [18], as endogenous estrogen seems to play a role in protecting women from CVD in their fertile period [6]. Moreover, glucose homeostasis is different in men and women [18]—women have lower fasting plasma glucose (FPG), higher 2 h plasma glucose following an oral glucose tolerance test (2-h PG) and a greater FPG—2-h PG increment. Compared to men, healthy women have lower skeletal muscle mass, higher adipose tissue mass, higher levels of circulating free fatty acids, and higher intramyocellular lipid content; all these factors promote higher insulin resistance in women [18,19]. An important factor in the differential impact of DM between men and women in various diseases is that treatment may differ according to gender. It has been widely reported that, after a cardiovascular event, women are undertreated in comparison with men [2,7]. Dyslipidemia is also less controlled in women than in men [6,20]. Increases in BMI over the individuals’ lifetime are greater in women than in men [3,6,20], and this higher BMI in women can contribute to the higher burden of diabetes among them [6]. Moreover, women have a different body fat distribution to men [2,7]. All the aforementioned factors can potentially contribute to the differences between cardiovascular risk in men and women, and gender-targeted treatment should eventually be considered [6]. 

The knowledge of these sex-related differences in the impact of DM in acute PE suggests the appropriateness of tailoring prevention strategies and clinical practices towards diabetes in PE. Sex-sensitive and culturally tailored prevention programs, as well as sex-specific education, lifestyle programs and drug therapy, will eventually contribute to a better care of patients with T2DM in the future, particularly for women, and also in the context of PE [18,19,20].

Few reports in the literature state rates of reperfusion therapy, and the rates reported are not uniform, ranging from around 3% [21] to 12% [22]. The thrombolysis rate in our cohort is somewhat elevated, however the fact that we excluded patients with concomitant active neoplasia may have contributed to gathering a group of patients more likely to be aggressively treated when necessary. In our patient cohort, hypertension history was associated with a survival advantage. This is intriguing and, in fact, we have no explanation for this finding. Hypertension has been proven to impart a survival advantage in some conditions, namely heart failure [23]. Such an unexpected association could be due to a reverse epidemiology phenomenon [24], and this should eventually be addressed in future studies, but is clearly beyond the scope of this study. The trend towards better survival among patients with central and bilateral acute PE is also intriguing and, again, we have no complete explanation for it. It is possible that these more extensive PE were diagnosed and treated earlier, however this explanation is speculative and beyond the scope of the study. 

The study has important limitations that are worth noting. It is a single center study with generalizability concerns. Furthermore, the retrospective design has associated problems related with the availability and quality of the data, namely, no complete data were available on important Pulmonary Embolism Severity Index (PESI) score variables, such as vital signs, at acute PE diagnosis; these might have had higher prognostic influence than gender or DM, and were not accounted for. The fact that patient selection was based on ICD9 coding makes under-reporting a possible concern. Over-reporting was prevented because all files were reviewed for diagnostic criteria, and patients with presumptive but non-confirmed PE were not included in the analysis, and also patients with recurrent PE were excluded from the study. Patients with concomitant active neoplasia were not included in the study because of the obvious interference with mortality, however it is important to keep in mind that the reported results are not applicable to a significant proportion of patients with acute PE, since neoplastic disease is a well-established risk factor for VTE [13,16]. An echocardiogram was not performed on a substantial proportion of patients (34%), and right ventricular dysfunction was not accounted for in the multivariate analysis. Troponin I was also not included because 24% patients had no available data. Moreover, the patients’ DM control in the acute condition and during follow up was not known and not included in the analysis, and that may have influenced the outcome. A better characterization of both the PE’s severity and the DM’s duration and control should be pursued in future studies.

Despite the reported limitations, we studied a large enough population of acute PE patients, for a long enough period, to detect the differential role of DM in mortality according to gender. Women with acute PE are more negatively affected by the coexistence of DM than men. Diabetic women with acute PE have a long-term death risk that is more than double that of non-diabetics, while DM does not appear to have such a negative impact on survival among men. As far as an extensive literature review could determine, this is the first study reporting such a differential impact of DM between genders, specifically in the PE context. Further studies should address the specific mechanisms behind this interaction between gender and DM in multiple diseases, and the reasons for women being more affected than men. This could ultimately lead to a different approach to DM depending on gender.

## 5. Conclusions

Gender influences the impact of DM on the survival of acute PE patients. DM worsens prognosis in women but not in men. A gender-tailored approach to DM in PE patients should be considered and investigated.

## Figures and Tables

**Figure 1 jcm-09-03511-f001:**
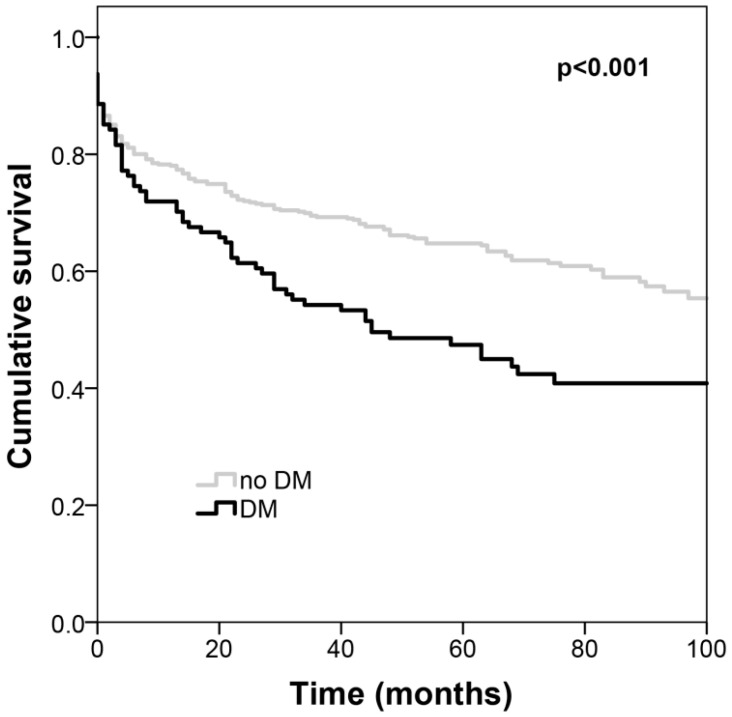
Kaplan–Meier survival curves of patients with acute PE according to the coexistence of DM. Diabetics have worse survival. PE: pulmonary embolism; DM: diabetes mellitus.

**Figure 2 jcm-09-03511-f002:**
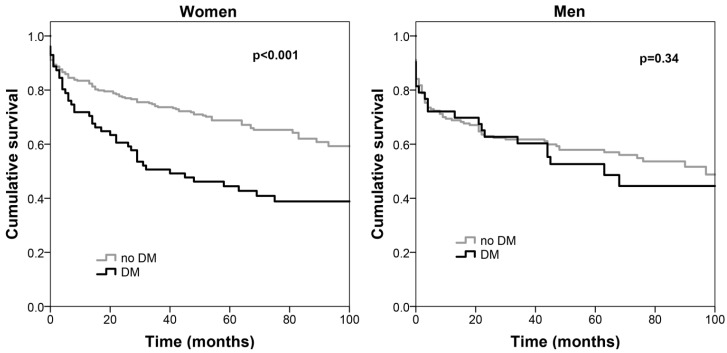
Kaplan–Meier survival curves according to the coexistence of DM separately in women (**left**) and men (**right**). Diabetes implies a survival disadvantage only in women. DM: diabetes mellitus.

**Table 1 jcm-09-03511-t001:** Patients’ characteristics and comparison between genders.

Characteristic	All (*n* = 577)	Women (*n* = 364)	Men (*n* = 213)	*p*-Value
**Male gender, *n* (%)**	213 (36.9)			
**Age (years), mean (SD)**	65 (19)	67 (18)	62 (19)	0.003
**Diabetes mellitus, *n* (%)**	114 (19.8)	71 (19.5)	43 (20.2)	0.84
**Arterial Hypertension, *n* (%)**	318 (55.1)	221 (60.7)	97 (45.5)	<0.001
**Haemoglobin (g/dL), mean (SD)**	12.5 (2.3)	12.1 (2.0)	13.1 (2.2)	<0.001
**Creatinine (mg/dL), median (IQR)**	0.93 (0.72–1.30)	0.90 (0.70–1.30)	1.00 (0.80–1.30)	0.003
**Lymphocytes (cells/µL), median (IQR)**	1600 (1015–2270)	1740 (1110–2340)	1410 (950–2030)	0.001
**Platelets**	205 (159–267)	220 (167–287)	187 (146–239)	<0.001
**C–Reactive Protein (mg/L), median (IQR)**	39.8 (15.7–93.7)	34.8 (14.2–79.6)	51.9 (17.7–115.4)	0.005
**BNP (pg/mL), median (IQR)**	283.0 (85.9–744.6)	284.8 (88.9–759.1)	278.8 (80.8–679.5)	0.87
**Central/bilateral PE, *n* (%)**	175 (30.4)	115 (31.8)	60 (28.2)	0.36
**idiopathic PE, *n* (%)**	376 (65.2)	237 (65.1)	139 (65.3)	0.97
**Thrombolytic therapy, *n* (%)**	67 (11.6)	45 (12.4)	22 (10.3)	0.46
**All–cause death, *n* (%)**	240 (41.6)	140 (38.5)	100 (46.9)	0.05

BNP: B type natriuretic peptide; IQR: interquartile range; PE: pulmonary embolism; SD: standard deviation.

**Table 2 jcm-09-03511-t002:** Survival predictors in acute PE: Cox-multivariate model.

Characteristics	HR (95% CI)	*p*-Value
**Diabetes mellitus**	2.33 (1.51–3.61)	<0.001
**Male gender**	1.84 (1.27–2.68)	0.001
**Arterial hypertension history**	0.67 (0.47–0.97)	0.03
**Central and bilateral Pulmonary embolism**	0.72 (0.51–1.02)	0.06
**C–reactive protein (per 10 mg/L)**	0.98 (0.96–1.01)	0.15
**B–type natriuretic peptide (per 100 pg/mL)**	1.02 (1.01–1.03)	<0.001
**Anemia**	1.22 (0.89–1.67)	0.23
**Creatinine >1.5 mg/dL**	1.76 (1.24–2.52)	0.002
**Lymphocyte counts <1500 cells/µL**	1.84 (1.34–2.53)	<0.001
**Platelets (×100)**	0.91 (0.76–1.09)	0.31
**Age >65 years**	3.06 (2.05–4.57)	<0.001
**Thrombolytic therapy**	0.73 (0.45–1.20)	0.22
**Interaction gender * Diabetes *mellitus***		0.04

CI: confidence interval, HR: hazard ratio. *: interaction between diabetes mellitus and gender.

**Table 3 jcm-09-03511-t003:** Survival predictors in acute PE. Multivariate analysis stratified by gender.

	Women		Men	
Characteristics	HR (95% CI)	*p*-Value	HR (95% CI)	*p*-Value
**Diabetes mellitus**	2.31 (1.45––3.65)	<0.001	1.10 (0.59–2.04)	0.77
**Arterial hypertension history**	0.64 (0.39–1.04)	0.07	0.75 (0.44–1.27)	0.29
**Central/bilateral PE**	0.72 (0.47–1.12)	0.15	0.74 (0.39–1.38)	0.33
**C–reactive protein (per 10 mg/L)**	0.98 (0.95–1.01)	0.22	1.00 (0.96–1.03)	0.81
**BNP (per 100 pg/mL)**	1.01 (1.00–1.03)	0.02	1.03 (1.01–1.04)	<0.001
**Anemia**	1.11 (0.73–1.67)	0.64	1.39 (0.82–2.35)	0.22
**Creatinine >1.5 mg/dL**	1.89 (1.20–2.99)	0.006	1.42 (0.76–2.66)	0.27
**Lymphocyte counts <1500 cell/µL**	1.87 (1.24–2.83)	0.003	1.62 (0.95–2.76)	0.08
**Platelets ×100**	0.91 (0.73–1.13)	0.41	0.85 (0.59–1.21))	0.36
**Age >65 years**	4.87 (2.57–9.26)	<0.001	2.20 (1.29–3.78)	0.004
**Thrombolytic therapy**	0.75 (0.41–1.37)	0.35	0.75 (0.30–1.88)	0.55

BNP: B-type natriuretic peptide; CI: confidence interval; HR: hazard ratio.

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
