# Peer review of "The Influence of Gender in The Prognostic Impact of Diabetes mellitus in acute Pulmonary Embolism"

_jcm, 2020, doi:10.3390/jcm9113511_

Round 1

Reviewer 1 Report

The Reviewer has read the manuscript entitled “The Influence of Gender in The Prognostic Impact of Diabetes mellitus in acute Pulmonary Embolism”. The authors perform a single-centre, retrospective study on PE patients and mainly stratify on gender and DM. They show that DM is a risk factor for all-cause mortality following acute PE, which appeared to be most pronounced in women as male diabetics did not have increased mortality.

In general, risk stratification is a central issue in acute PE as higher risk help dictate therapeutically approach or at least level of surveillance of an admitted patient. This is mainly a concern for the intermediate-risk group according to the ESC 2019 guidelines. 

The study does have some major as well as minor concerns to be adressed:

Major:

  1. Introduction: several statements on gender-differences in DM lack references.

  1. Introduction: Main statement for the aim of the study is, that DM increases mortality in PE. The statement refers to ref 14; however, in that paper, PE or VTE is not mentioned. Please correct.

  1. Methods/Results: the level of details on the two major focus areas (DM and PE) is not sufficient. On PE patients; hemodynamic status of patients? Troponins, echocardiography, RV/LV from diagnostic CT scan? Vitals at presentation? Those would provide insight in the severity of the patients and put perspective on the added value of DM and gender. And for DM patients: what was their HbA1c? Was it Insulin-dependent vs not, which is important distinction in other cardiovascular diseases? What was the number of anti-diabetic drugs? The study could explore on the severity and type of DM vs its impact on acute PE patients.

  1. Results: Finding of diabetes as prognostic marker with HR 2.3 corresponds well to present knowledge in the ESC 2019 PE guidelines. Is the novelty that this is mainly in women? Should be clearer.

Minor:

  1. Abstract: please include data on main findings (DM all-cause mortality vs non-DM HR2.31 and HR for women). Is the two last sentences the conclusion very similar?
  2. Introduction: Language can be improved, as well as content without repetitions e.g. line 38-47. Line 49-52 completely unnecessary for the present study, due to many oncological authors?
  3. Introduction can be more direct towards the topic, e.g. DM is frequent and has gender-related differences -> VTE is frequent -> DM is a poor prognostic marker in PE. Aim:…
  4. Introduction: Please state clear hypothesis if present prior to study completion. Why reference on an aim?
  5. Methods: Improve language without repetitions e.g. line 63-64 was just stated before. Please use more logical/chronological order, e.g. demographic data from admission prior to follow-up. Consider sub-headings.
  6. Methods: Excluded patients with no imagological confirmation but includes on V/Q. V/Q is not a confirmed diagnosis but mostly interpreted if “high probability”. Please clarify.
  7. Results:
  8. Tabel 1: How is idiopathic PE defined? Why include lymphocytes but not thrombocytes which is relevant in acute PE?
  9. Multivariate analysis could include more relevant parameters than anemia and lymphopenia, i.e. characterization of PE and DM patients.
  10. 11.6% thrombolysis is very high, please comment.
  11. All-cause death until 2017 or? Please clarify and state range of follow-up.
  12. Write 95%CI and not IC 95%
  13. Male sex is risk factor, hypertension protective, please comment. Central/bilateral and thrombolysis trend to protective, please comment.
  14. Correct spelling mistakes in both tables.
  15. Discussion: Discussing mechanisms lines 141-170 lack perspective to PE. Please focus the discussion to the aim of the study.

Author Response

Dear editors of the Journal of Clinical Medicine,

We thank you for the opportunity to revise the manuscript: jcm-946811 entitled “The influence of gender in the prognostic impact of Diabetes mellitus in acute pulmonary embolism”. We have altered the manuscript according to the reviewer’s suggestions in order to improve it. We hope that, in its revised version, you find it worth for publication in the Journal of Clinical Medicine. Please let us know if any more alterations are necessary.

Reviewers’ comments

Reviewer 1

The study does have some major as well as minor concerns to be addressed:

Major:

  1. Introduction: several statements on gender-differences in DM lack references.

We totally agree with the reviewer and, in fact, some statements needed references. References were added as requested. See new and revised references:

  1. Wang, Hao Ba, Ying Cai, Run-Ce Xing, Qian. (2019). Association between diabetes mellitus and the risk for major cardiovascular outcomes and all-cause mortality in women compared with men: a meta-analysis of prospective cohort studies. BMJ Open. 9.
  2. Wang, Yafeng O'neil, Adrienne Jiao, Yurui Wang, Lijun Huang, Jingxin Lan, Yutao Zhu, Yikun Yu, Chuanhua. (2019). Sex differences in the association between diabetes and risk of cardiovascular disease, cancer, and all-cause and cause-specific mortality: A systematic review and meta-analysis of 5,162,654 participants. BMC Medicine. 17.
  3. Elling, Devy Surkan, Pamela Enayati, Sahba El-Khatib, Ziad. (2018). Sex differences and risk factors for diabetes mellitus - An international study from 193 countries. Globalization and Health.14.
  4. Alfredsson, Joakim Green, Jennifer Stevens, Susanna Reed, Shelby Armstrong, Paul Bethel, M. Engel, Samuel McGuire, Darren Van de werf, Frans Hramiak, Irene White, Harvey Peterson, Eric Holman, Rury. (2018). Sex differences in management and outcomes of patients with type 2 diabetes and cardiovascular disease: A report from TECOS. Diabetes, Obesity and Metabolism. 20.
  5. Nataraj N, Ivy JS, Payton FC, Norman J. Diabetes and the hospitalized patient: A cluster analytic framework for characterizing the role of sex, race and comorbidity from 2006 to 2011. Health Care Manag Sci. 2018
  6. Peters, Sanne Muntner, Paul Woodward, Mark. (2019). Sex Differences in the Prevalence of, and Trends in, Cardiovascular Risk Factors, Treatment, and Control in the United States, 2001 to 2016. Circulation. 139.
  7. Peters SA, Huxley RR, Woodward M. Diabetes as risk factor for incident coronary heart disease in women compared with men: a systematic review and meta-analysis of 64 cohorts including 858,507 individuals and 28,203 coronary events. Diabetologia. 2014 Aug;57(8):1542-51. doi: 10.1007/s00125-014-3260-6. Epub 2014 May 25. PMID: 24859435.
  8. Maric-Bilkan, Christine. (2017). Sex differences in micro- and macro-vascular complications of diabetes mellitus. Clinical Science. 131.
  9. Patricia, Palau Bertomeu-González, Vicente Sanchis, Juan Soler, Meritxell de la Espriella-Juan, Rafael Domínguez, Eloy Santas, Enrique Núñez, Eduardo Chorro, Francisco Miñana, Gema Bayes-Genis, Antoni Nuñez Villota, Julio. (2019). Differential prognostic impact of type 2 diabetes mellitus in women and men with heart failure with preserved ejection fraction. Revista Española de Cardiología (English Edition). 73.
  10. Ohkuma, Toshiaki Komorita, Yuji Peters, Sanne Woodward, Mark. (2019). Diabetes as a risk factor for heart failure in women and men: a systematic review and meta-analysis of 47 cohorts including 12 million individuals. Diabetologia. 62.

  1. Introduction: Main statement for the aim of the study is, that DM increases mortality in PE. The statement refers to ref 14; however, in that paper, PE or VTE is not mentioned. Please correct.

The aim of the study was to investigate if there were gender related differences in the prognostic impact of DM in PE such as it has been reported for other cardiovascular conditions. The statement that DM is a predictor of poor outcome in PE is obviously followed by an incorrect reference. We apologize for the mistake. In the current version the reference has been altered to de Miguel-Díez J, López-de-Andrés A, Jiménez-Trujillo I, Hernández-Barrera V, Jiménez-García R, Lorenzo A, et al. Mortality after pulmonary embolism in patients with diabetes. Findings from the RIETE registry. Eur J Intern Med. 2019;59:46-52.

  1. Methods/Results: the level of details on the two major focus areas (DM and PE) is not sufficient. On PE patients; hemodynamic status of patients? Troponins, echocardiography, RV/LV from diagnostic CT scan? Vitals at presentation? Those would provide insight in the severity of the patients and put perspective on the added value of DM and gender. And for DM patients: what was their HbA1c? Was it Insulin-dependent vs not, which is important distinction in other cardiovascular diseases? What was the number of anti-diabetic drugs? The study could explore on the severity and type of DM vs its impact on acute PE patients.

This is a retrospective study with problems mainly concerning data availability. These are, of course, very pertinent questions, however we do not have enough data to perform such a multivariate analysis. 34% of the patients with acute PE diagnosis had no echocardiogram performed during hospitalization and 24% of the them had no TpI measurement. Not all patients were admitted due to acute PE, in some of them this event was a complication of the hospitalization or the diagnosis was only made later. We have data on admission hemodynamic status but not from the moment in which the diagnosis was made. The acute PE severity is mainly inferred from the thrombolysis need.

Also, there were no available data to characterize DM in most of the 19.8% diabetics.This is a major setback of the study and we now state it clearly in the discussion section. However the aim of the study was not to assess the impact of the severity of DM in its prognostic impact on acute PE patients, but to study a possible interaction of gender in the prognostic impact of DM in acute PE. With such a small sample size and a retrospective design many questions have to remain unanswered.

We did, however, and as suggested in a later question from this reviewer, included the platelet counts in the multivariate analysis. Also, if TpI and right ventricular dysfunction were included in the model, results would be similar but little more than 300 patients would be include in the analysis. 

Please see the added sentences in the discussion section: An echocardiogram was not performed in a substantial proportion of patients (34%) and right ventricular dysfunction was not accounted for in the multivariate analysis. Troponin I was also not included because many patients had no available data. Moreover, patients’ DM control in the acute condition and during follow up was not known and not included in the analysis and that may have influenced the outcome. A better characterization of both the PE severity and of the DM duration and control should be pursued in future studies.” (lines 161-166)

  1. Results: Finding of diabetes as prognostic marker with HR 2.3 corresponds well to present knowledge in the ESC 2019 PE guidelines. Is the novelty that this is mainly in women? Should be clearer.

The novelty is, in fact, that the prognostic impact of DM in acute PE is mainly in women. We recognize in the introduction section that DM is a known predictor of poor outcome in PE –please see “PE patients with DM have been shown to have higher mortality than non-diabetics” (lines 48-49).

In the results section we start by reporting that Among acute PE patients those with DM presented higher all-cause mortality with aHR=2.33 (95% CI 1.51-3.61) when compared with non-diabetics;...” (please see lines 102-104), but later we say that “When analysis was stratified according to gender, DM was an independent predictor of all-cause mortality in patients with acute PE only in women - HR= 2.31 (95%CI 1.45-3.65), p<0.001” (lines 107-100). We therefore start by reproducing what is already known for the whole patient population. We then stratify the analysis according to gender to study the interaction between DM and gender and verify that the association is only reproduced in women and that the impact of DM in women with acute PE is significantly different than the one of DM in men with PE. The novelty of the manuscript is later reinforced in the conclusions – “Gender influences the impact of DM on survival of acute PE patients. DM worsens prognosis in women but not in men.” (lines 177-178).

Also in the abstract the novelty is clearly stated – please see the 2 last sentences of the abstract “Diabetic women with PE have twice the long-term mortality risk; DM is not mortality-associated in men”. (lines 26, 27)

Minor:

1.Abstract: please include data on main findings (DM all-cause mortality vs non-DM HR2.31 and HR for women). Is the two last sentences the conclusion very similar?

According to the reviewer’s idea these data were included in the abstract.Also as suggested, and due to similarity, the last sentence was removed since it was implied in the one that remains.

The alterations in the abstract are as follows “…Diabetics presented higher all-cause mortality: Hazard ratio (HR)=2.33 (95% confidence Interval (CI): 1.51-3.61) when compared with non-diabetics. However, when analysis was stratified according to gender, DM was independently associated with worse prognosis only in women: HR= 2.31 (95%CI 1.45-3.65), while in men, the HR was 1.10 (95% CI 0.59-2.04). Interaction between gender and DM was significant (p=0.04). Gender influences the prognostic impact of DM in acute PE. (please see lines 22-26)

2.Introduction: Language can be improved, as well as content without repetitions e.g. line 38-47. Line 49-52 completely unnecessary for the present study, due to many oncological authors?

  1. Introduction can be more direct towards the topic, e.g. DM is frequent and has gender-related differences -> VTE is frequent -> DM is a poor prognostic marker in PE. Aim:…

The introduction has been altered according to the reviewer’s suggestions – please see answers to previous question (3) and next question (5). We think that in its current version it is more direct towards the topic. We start by saying that DM is frequent and is a poor prognostic predictor in many cardiovascular setting (1st paragraph of the introduction), we then explore the gender related differences in diabetes and comorbidities/prognosis (2nd paragraph). In the 3rd paragraph we say that VTE is frequent and that DM predicts ominous outcome in PE and, in the last paragraph we state our aim and our pre-specified hypothesis.

  1. Introduction: Please state clear hypothesis if present prior to study completion. Why reference on an aim?

  1. Methods: Improve language without repetitions e.g. line 63-64 was just stated before. Please use more logical/chronological order, e.g. demographic data from admission prior to follow-up. Consider sub-headings.

The chronological order was, in fact, neither appropriate nor logical. We have therefore altered the order following the reviewers suggestion (please see in the text). The explanation of the endpoint and follow-up are now after demographic data, admission data and comorbidities definitions. We have also deleted the suggested lines to avoid unnecessary repetitions

  1. Methods: Excluded patients with no imagological confirmation but includes on V/Q. V/Q is not a confirmed diagnosis but mostly interpreted if “high probability”. Please clarify.

The majority of patients had their diagnosis confirmed by CTPA, however, a small percentage of them (in which there was contraindication to CTPA) the diagnosis was considered when the nuclear ventilation-perfusion (V/Q) imaging was classified as has high probability. This is in accordance with the most recent 2019 ESC acute PE guidelines that consider as a recommendation class IIa, level of evidence B that “It should be considered to accept that the diagnosis of PE (without further testing) if the V/Q scan yields high probability for PE.” (table 4.11 recommendations for diagnosis; page 559).

With considered that a V/Q scan classified as of “high probability” was implicit in the confirmed diagnosis of PE, however we have made the sentence in the methods more clear – please see in page 3, lines 59 and 60 All patients had acute PE confirmed by computed tomography pulmonary angiography (CTPA) or by a high probability nuclear ventilation-perfusion (V/Q) imaging.”

  1. Results:
  2. Table 1: How is idiopathic PE defined? Why include lymphocytes but not thrombocytes which is relevant in acute PE?

The classification of acute PE as idiopathic or provoked was according to the attending physician. When physicians' information was not available (approximately 5%) patients were classified as having idiopathic or provoked according to the opinion of two experienced internists. PE was considered idiopathic when no neoplastic condition or major condition favoring embolism was identified. Patients with a know or newly diagnosed neoplastic condition were excluded from the analysis. The investigation and therapeutic approach was at the discretion of the attending physician and this may have led to an underestimation of the number of patients with PE secondary to neoplasia and non-neoplastic condition since there is lack of a uniform protocol and lack of consensus concerning which is the best diagnostic/investigation workup in such cases. Therefore, the complete screening for both neoplasia and coagulation disturbances and thrombophilia was not guaranteed.

According to the reviewers’ suggestion we have also included platelet counts and, despite differences between men and women they had no prognostic impact in patients with acute PE. Maybe this is not surprising because platelets are mainly related with arterial and not venous thrombotic phenomena.

Please see new tables 1,2 and 3 with inclusion of the variable: platelets.

  1. Multivariate analysis could include more relevant parameters than anemia and lymphopenia, i.e. characterization of PE and DM patients.

Please see answer to major question 3 and minor question 8.

Data on platelet counts were included in the comparison between men and women and also in the multivariate models. New tables are provided however results are basically similar.

This is a retrospective study with problems mainly concerning data availability.

  1. 11.6% thrombolysis is very high, please comment.

In the literature, there are not many reports stating rates of reperfusion therapy, and rates reported are not uniform. In BMJ 2019;366:l4416 http://dx.doi.org/10.1136/bmj.l4416 we can read that “Regarding in-hospital reperfusion treatments, patients at low volume centers were more likely to receive reperfusion treatments (mostly systemic thrombolysis; 3.9% v 3.0%, P<0.001); and in J Geriatr Cardiol 2020; 17: 510-518. doi:10.11909/j.issn.1671-5411.2020.08.005 we can read “…19% of patients were classified as high-risk PEs. Within this cohort of 134 patients, 12% of patients underwent CDT with recombinant tissue plasminogen activator (rt-PA), 5% of patients received full dose (rt-PA: 100 mg), 13% of patients were treated with systemic half-dose (rt-PA: 50 mg), 5% of patients underwent surgical embolectomy and 3% of patients mechanical thrombectomy.”

Our reported rate of thrombolysis was intermediate between these two studies. However, the fact that we excluded patients with concomitant active neoplasia may have contributed to gather a group of patients more likely to be aggressively treated when necessary.

  1. All-cause death until 2017 or? Please clarify and state range of follow-up.

We have added such information in the results section. Please see lines 98 and 99: Patients were followed for a median (interquartile range (IQR) period of 52 (12-76) months.”

  1. Write 95%CI and not IC 95%

Corrected.

  1. Male sex is risk factor, hypertension protective, please comment. Central/bilateral and thrombolysis trend to protective, please comment.

Male gender is part of the original Pulmonary Embolism Severity Index (PESI). Males score immediately 10 points. Despite this, the majority of studies done on gender differences on pulmonary embolism encounter a higher mortality rate in women and this is still open to research. For example in the Journal of thrombosis and thrombolysis (2015),Simplified PESI score and sex difference in prognosis of acute pulmonary embolism: a brief report from a real life study, Masotti Luca et all,conclude that “.. in real life in patients with acute PE different co-morbidity burdens in females compared to males… Females are likely to have advanced age but unlikely to have cancer and cardio-respiratory diseases compared to males. Females are at significantly lower risk of all cause in-hospital mortality for sPESI score B2 (?) but at higher risk of bleeding, irrespective from sPESI scoring. The ability of sPESI to predict all-cause mortality seems better in females. Further prospective studies aimed to better clarify sex difference in prognostic assessment of acute PE are warranted.”

Hypertension is not part of the PESI score. Non-survivors have higher prevalence of hypertension history than survivors (50.4 vs 61.7%, respectively, p value=0.008) and in a univariate approach hypertension history is also predictive of ominous outcome with an HR of 1.35 (1.04-1.75), p=0.02.

When DM and gender are included in the model the association with mortality is lost and in the final model, hypertension history appears as a protective factor.

Variables in the Equation

B

SE

Wald

df

Sig.

Exp(B)

95.0% CI for Exp(B)

Lower

Upper

DM

.406

.154

6.993

1

.008

1.501

1.111

2.028

Male gender

.295

.132

5.001

1

.025

1.343

1.037

1.739

Hypertension

.228

.141

2.604

1

.107

1.256

.952

1.655

Variables in the Equation

B

SE

Wald

Df

Sig.

Exp(B)

95.0% CI for Exp(B)

Lower

Upper

DM

.847

.223

14.453

1

.000

2.333

1.507

3.610

Male gender

.611

.190

10.310

1

.001

1.843

1.269

2.677

Hypertension

-.395

.184

4.627

1

.031

.674

.470

.966

PE_central_bilateral

-.329

.178

3.444

1

.063

.719

.508

1.019

RCPx10

-.017

.012

2.103

1

.147

.983

.961

1.006

BNPX100

.019

.004

17.982

1

.000

1.019

1.010

1.028

Anemia

.197

.162

1.470

1

.225

1.217

.886

1.673

creat>1.5

.567

.181

9.817

1

.002

1.763

1.237

2.515

lymphopenia<1500

.608

.162

14.017

1

.000

1.837

1.336

2.526

PlaletelsX100

-.094

.094

1.018

1

.313

.910

.758

1.093

Age>65

1.118

.205

29.846

1

.000

3.058

2.048

4.567

Thrombolysis

-.309

.251

1.518

1

.218

.734

.449

1.200

DM*Gender

-.732

.355

4.254

1

.039

.481

.240

.964

This is intriguing and, in fact, we have no explanation to this finding. Hypertension has been proved to be a survival advantage in many conditions namely heart failure and this should eventually be addressed in future studies but is clearly beyond the scope of the study.

Concerning the extension and location of the acute PE, survivors tend to have higher prevalence of central and bilateral acute PE (34.2% against 25.1% in non-survivors).

The univariate Cox-regression analysis supports this finding.

Variables in the Equation

B

SE

Wald

df

Sig.

Exp(B)

95.0% CI for Exp(B)

Lower

Upper

PE_Central_Bilateral

-.385

.150

6.610

1

.010

.680

.507

.913

And in the multivariate approach a trend towards survival advantage is seen.

This is an also intriguing finding and again we have no complete explanation for it. The fact that patients with a neoplastic active condition in which the threshold for acute PE suspicion is lower and the diagnosis is probably made earlier when the condition is not yet so established may have contributed to an over-expression of such cases. Another possible explanation is the fact that maybe this more extensive acute PE were diagnosed and treated earlier. These arguments are, of course, speculative and beyond the scope of the study. However it is important to remember that in the multivariate approach statistical significance was not reached.

  1. Correct spelling mistakes in both tables.

Spelling mistakes in the tables have been corrected.

  1. Discussion: Discussing mechanisms lines 141-170 lack perspective to PE. Please focus the discussion to the aim of the study.

We completely agree with your suggestion. We have altered the discussion in accordance and introduced some sentences in order to contextualize the issue. The idea behind our explanation is that VTE shares risk factors and inflammatory mechanisms with other cardiovascular diseases. We try to do a parallelism with these diseases (coronary heart disease, pro-thrombotic mechanism) by further explaining them. It is not easy to focus the discussion because there are not many studies of gender differences specifically in VTE. A parallelism with other cardiovascular diseases is made in order to make suggestions and draw hypothesis behind the differences found. Please review mainly lines 125-128, 161-165.

Reviewer 2 Report

The aim of the study was to assess the influence of gender in the prognostic impact of diabetes mellitus in pulmonary embolism.
The authors stated that gender influences the impact of diabetes mellitus on survival of acute pulmonary embolism, in fact women showed a worse prognosis in their cohort. 

The study presented interesting findings and methodologically is well conducted, so I would suggest only minor changes. 

  1. In the abstract the abbreviations need to be preceded (the first time) by a full text explanation. 
  2. Line 44. Citation number needs to be changed, according to the journal style.
  3. Line 76. Instead of abstracted maybe is better extracted. 
  4. Line 92. Please specify which cut-off did you use to choose the variable to put in the multivariate model. p<0.05? 
  5. Line 107. There is a discrepancy betweeen 41.5% and the table 1 in which percentage of death is 41.6%.
  6. Line 154. Instead of genre maybe is better gender.
  7. I think it would be useful to put a table (even in supplemental material) with in the column survivors vs. not survivors.  

Author Response

Dear editors of the Journal of Clinical Medicine,

We thank you for the opportunity to revise the manuscript: jcm-946811 entitled “The influence of gender in the prognostic impact of Diabetes mellitus in acute pulmonary embolism”. We have altered the manuscript according to the reviewer’s suggestions in order to improve it. We hope that, in its revised version, you find it worth for publication in the Journal of Clinical Medicine. Please let us know if any more alterations are necessary.

Reviewers’ comments

Reviewer 2

The aim of the study was to assess the influence of gender in the prognostic impact of diabetes mellitus in pulmonary embolism.
The authors stated that gender influences the impact of diabetes mellitus on survival of acute pulmonary embolism, in fact women showed a worse prognosis in their cohort. 

The study presented interesting findings and methodologically is well conducted, so I would suggest only minor changes. 

  1. In the abstract the abbreviations need to be preceded (the first time) by a full text explanation. 

According to the reviewers request we have provided a full text explanation of the abbreviations – DM, PE, BNP, HR and CI.

  1. Line 44. Citation number needs to be changed, according to the journal style.

The layout was changed according to journal style.

  1. Line 76. Instead of abstracted maybe is better extracted. 

  1. Line 92. Please specify which cut-off did you use to choose the variable to put in the multivariate model. p<0.05? 

Yes, the cut-off of 0.05 for p value was considered as a statistically significant difference between groups (men vs women). That is now more clearly explained in the subsection statistical analysis. Please see The prognostic impact of DM in acute PE was assessed using a Cox-regression analysis. A multivariate model was built based on variables differently distributed in men and women, p value<0.05 was considered as statistically significant.”, lines 84-86

  1. Line 107. There is a discrepancy between 41.5% and the table 1 in which percentage of death is 41.6%.

The correct percentage is 41.6% (240/577=41.59%). The discrepancy has been corrected and the percentage in the text has been changed to 41.6%.

  1. Line 154. Instead of genre maybe is better gender.

As suggested, the mistaken word was altered to “gender”.

  1. I think it would be useful to put a table (even in supplemental material) with in the column survivors vs. not survivors.  

As suggested by the reviewers we now provide a table with the comparison between survivors and non-survivors as supplementary material. Please see the added sentence in the first paragraph of the results section – “Comparison between survivors and non-survivors is provided as supplementary material (supplementary table 1)” (lines 100 and 101).

See also the added table

Table 1S. Comparison between survivors and non-survivors"

Characteristic

Survivors (n=337)

Non-survivors (240)

p-value

Male gender, n (%)

113 (33.5)

100 (51.7)

0.05

Age (years), mean  (SD)

59 (18)

75 (14)

<0.001

Diabetes mellitus, n (%)

49 (14.5)

65 (27.1)

<0.001

Arterial Hypertension, n (%)

170 (50.4)

148 (61.7)

0.08

Haemoglobin (g/dL), mean (SD)

12.6 (2.2)

12.3 (2.3)

0.15

Creatinine (mg/dL), median (IQR)

0.84 (0.70-1.10)

1.13 (0.87-1.66)

<0.001

Lymphocytes (cells/µL), median (IQR)

1780 (1200-2350)

1320 (848-1975)

<0.001

Platelets

213 (167-274)

190 (152-259)

0.01

C-Reactive Protein (mg/L), median (IQR)

38.7 (14.2-85.5)

42.0 (18.5-102.1)

0.13

BNP (pg/mL), median (IQR)

180.1 (49.1-450.5)

438.2 (173.5-1024.3)

<0.001

Central/bilateral PE, n (%)

115 (34.2)

60 (25.1)

0.02

Thrombolytic therapy, n (%)

44 (13.1)

23 (9.6)

0.20

Round 2

Reviewer 1 Report

Reviewer’s reply

Revised manuscript

I have read the revised manuscript entitled “The influence of gender in the prognostic impact of diabetes mellitus in acute pulmonary embolism”.

The manuscript in general has improved substantially due to the proposed changes.

In this response, the Reviewer will adhere to the numbers in the original comments. If numbers are removed, I believe the authors have responded appropriately.

Reviewer 1

The study does have some major as well as minor concerns to be addressed:

Major:

3. Methods/Results: the level of details on the two major focus areas (DM and PE) is not sufficient. I still find this a major limitation to be addressed more clearly in the limitation section.. Please change “many” in line 163 to 24% since it is a known number. Consider to include the analysis of the subset of patients with echocardiogram and/or troponins. The authors state it includes around 300 patients and might be a minor add to the Results saying “in patients with RV strain (defined as…) the HR was X versus those without RV strain”? And stratified by gender and/or DM status. The risk stratification on RV function is important in acute PE

Minor:

9. Multivariate analysis could include more relevant parameters than anemia and lymphopenia, i.e. characterization of PE and DM patients.

Authors reply: “This is a retrospective study with problems mainly concerning data availability.” The reviewer agree and accept so, but this limitation must be mentioned in the Limitations section e.g. after line 154; no data on e.g. PESI variables like vitals that might have higher prognostic influence than DM.

10. 11.6% thrombolysis is very high, please comment.

The authors cite 3% thrombolysis in one paper and 5-13% OUT OF 19% high-risk patients. The 11.6% thrombolysis rate of all patients still appears high. Consider commenting in the paper.

12. Write 95%CI and not IC 95%

Missed one in line 103

13. Male sex is risk factor, hypertension protective, please comment. Central/bilateral and thrombolysis trend to protective, please comment. New comment: the reviewer agree that the findings on hypertension and bilateral/central PE is suprising, consider commenting in the paper. Hypertension to be protective might be a result of absence of hypotension which is a high risk factor.

Author Response

Dear editors of the Journal of Clinical Medicine, we thank you for the opportunity to revise the manuscript: jcm-946811 entitled “The influence of gender in the prognostic impact of Diabetes mellitus in acute pulmonary embolism”. We have altered the manuscript according to the reviewer’s suggestions in order to improve it. We hope that, in its revised version, you find it worth for publication in the Journal of Clinical Medicine. Please let us know if any more alterations are necessary.

Reviewers’ comments

Reviewer 1

The study does have some major as well as minor concerns to be addressed:

Major:

  1. Methods/Results: the level of details on the two major focus areas (DM and PE) is not sufficient. I still find this a major limitation to be addressed more clearly in the limitation section. Please change “many” in line 163 to 24% since it is a known number. Consider to include the analysis of the subset of patients with echocardiogram and/or troponins. The authors state it includes around 300 patients and might be a minor add to the Results saying “in patients with RV strain (defined as…) the HR was X versus those without RV strain”? And stratified by gender and/or DM status. The risk stratification on RV function is important in acute PE

We have changed many to the know percentage as indicated by the reviewer.

We completely agree with the reviewer that right ventricular dysfunction is a very important variable, however with the 282 patients with all data available (including echocardiogram and TpI) we would be only analysing 105 events: 45 in men and 60 in women which is completely incompatible with a multivariate analysis with 13 variables (DM, arterial hypertension, central and bilateral TE, CRP, BNP, anaemia, creatinine, lymphopenia, platelets, age, thrombolysis, Troponin I and right ventricular dysfunction). We strongly advised the non presentation of those data because the risk of overfitting is elevated and it is statistically incorrect. Unfortunately, it is impossible to meet this request of the reviewer. The severity of acute PE is indirectly given by reperfusion therapy need.

These limitations - lack of echocardiogram data, Troponin I measurement and lack of vital signs - are now clearly stated in the limitations section.

Minor:

  1. Multivariate analysis could include more relevant parameters than anemia and lymphopenia, i.e. characterization of PE and DM patients.

Authors reply: “This is a retrospective study with problems mainly concerning data availability.” The reviewer agree and accept so, but this limitation must be mentioned in the Limitations section e.g. after line 154; no data on e.g. PESI variables like vitals that might have higher prognostic influence than DM.

Following the reviewers’ suggestion that limitation is now clearly stated. Please see after mentioned lines 158-160“...namely no complete data were available on important Pulmonary Embolism Severity Index (PESI) score variables like vital signs at acute PE diagnosis; these, might have had higher prognostic influence than gender or DM and were not accounted for.”

  1. 11.6% thrombolysis is very high, please comment.

The authors cite 3% thrombolysis in one paper and 5-13% OUT OF 19% high-risk patients. The 11.6% thrombolysis rate of all patients still appears high.

Consider commenting in the paper.

As suggested by the reviewer the elevated thrombolysis rate is now commented in the paper. Please see the added sentence in the discussion section: Few reports in the literature state rates of reperfusion therapy and rates reported are not uniform ranging from around 3% [21] to 12% [22]. The thrombolysis rate in our cohort is somewhat elevated, however, the fact that we excluded patients with concomitant active neoplasia may have contributed to gather a group of patients more likely to be aggressively treated when necessary.”Please see references added to the article.

  1. Write 95%CI and not IC 95%

Missed one in line 103

We apologize. The misspelling is corrected in the new version of the manuscript.

  1. Male sex is risk factor, hypertension protective, please comment. Central/bilateral and thrombolysis trend to protective, please comment. New comment: the reviewer agrees that the findings on hypertension and bilateral/central PE is surprising, consider commenting in the paper. Hypertension to be protective might be a result of absence of hypotension which is a high risk factor. 

Following the reviewers suggestion these intriguing findings are now commented in the manuscript. Please see the sentences that were added to the discussion section. Hypertension has been proved to portend survival advantage in some conditions namely heart failure [23]. Such unexpected association could be due to a reverse epidemiology phenomenon [24] and this should eventually be addressed in future studies but is clearly beyond the scope of the study. The trend towards better survival among patients with central and bilateral acute PE is also intriguing and, again, we have no complete explanation for it. It is possible that these more extensive PE were diagnosed and treated earlier, however, this explanation is speculative and beyond the scope of the study.”

We are not completely sure that we can say, as the reviewer suggest, that Hypertension to be protective might be a result of absence of hypotension which is a high risk factor, since there was no absence of hypotension – 11.6% of thrombolysis means that hypotension was a reality.